# OpenReview forum: "Monet: Mixture of Monosemantic Experts for Transformers"
_ICLR.cc/2025/Conference — ICLR 2025 Poster_

### Official Review · Reviewer_YJRi · 2024-10-28

**Soundness:** 3
**Presentation:** 2
**Contribution:** 2
**Rating:** 6
**Confidence:** 3

**Summary:**

In this paper, the authors propose Monet. A new sMoE achitecture built on top of PEER. By pushing the notation of expert to the limit, Monet shows superior performance and unique ability to unlearn domain knowledge by simply masking out experts.  Further analyses demonstrate mutual exclusivity of knowledge across experts and showcase the parametric knowledge encapsulated within individual experts.

**Strengths:**

* Simple and straightforward idea
* The experiments on domain masking and unlearning is interesting

**Weaknesses:**

* Presentation can be greatly improved. For example, figure 1 does more confusing than explaining. There is zero followup in the caption telling the readers what "E1", "BL2", "TL2" are. Because they are arbitrary abbreviation defined by the authors, they should be properly annotated, or simply just use the full name.
* No proper ablations to study different choices in the architectural design and no insight is provided. For example, can we mix Horizontal Expert Decomposition and vertical expert decomposition? Which part of the changes over PEER make it superior?
* No baseline comparison against PEER and traditional SMoE. How come these two most obvious baselines are missing?

Some other minor issues:
*  citation to PEER is missing.
*  Incremental proposal on top of PEER, I am uncertain how significant the contributions are

**Questions:**

* What does the model start with in table 3?

---

> ### Author Response · Authors · 2024-11-21
> **Response to Reviewer YJRi (Part 1/3)**
>
> We appreciate the comments about the method and are sorry about the confusion.
>
> > **Q1**. Presentation can be greatly improved.  \
> **Q2**. Which part of the changes over PEER make it superior?  \
> **Q3**. Incremental proposal on top of PEER, I am uncertain how significant the contributions are.
> >
>
> **Please refer to our updated manuscript**, where we have improved the readability in sections `2. Preliminaries` to through `3. Monet`. Based on your feedback **Q1**, we have also enhanced the presentations of `Figure 1` in the revision.
>
> To summarize sections 2 and 3:
>
> 1. Inspired by the product key algorithm [1], PEER [2] processes up to a million experts with product key retrieval.
> 2. Despite its computational efficiency, PEER requires to initialize and store $N$ standalone experts, resulting in memory usage that grows linearly with the number of experts, $O(N)$.
> 3. In response to **Q2**, our contribution is partitioning the expert’s MLP network into two different groups of segments and storing them within $O(\sqrt{N})$ memory constraint. During the training or inference, the learned router dynamically composes expert networks to form $N$ combinations of experts.
>
> Below is a comparison of time complexity for expert retrieval and space complexity for expert parameters:
>
> | **Model** | **Time Complexity** | **Space Complexity** |
> | --- | --- | --- |
> | SMoE | $O(Nd)$ | $O(Nmd)$ |
> | PEER | $O((\sqrt{N} + k^2)Hd)$ | $O(Nd)$ |
> | Monet | $O(\sqrt{N}Hd)$ | **$O(\sqrt{N}md)$** |
>
> where $d$ is the hidden dimension of the expert, $m$ is the dimension of the individual expert, $k$ is the TopK hyperparameter, and $H$ denotes multi-head of the router.
>
> Regarding **Q3**, we suggest that our contribution is significant considering that our product key composition has optimized space complexity while maintaining the time complexity of PEER.

---

> ### Author Response · Authors · 2024-11-21
> **Response to Reviewer YJRi (Part 2/3)**
>
> > No baseline comparison against PEER and traditional SMoE.
>
> Following your request for a traditional SMoE interpretability baseline, we have included knowledge unlearning of OLMoE [3]. OLMoE LLM with total 6.9B parameters has been selected as the representative baseline for conventional SMoE architectures for two reasons: (1) it has the largest number of experts among the publicly available SMoE LLMs [3-5] and (2) it has been trained with an extensive amount of tokens from various sources.
>
> **Monet-VD 1.4B’s Domain Masking Performance Perturbation in MMLU**
>
> ||biology|business|chemistry|compsci|economics|engineering|health|history|law|math|other|philosophy|physics|psychology|
> |-|:-:|:-:|:-:|:-:|:-:|:-:|:-:|:-:|:-:|:-:|:-:|:-:|:-:|:-:|
> |$\Delta$ Target Domain|-4.66|-4.61|-5.49|-1.05|-2.32|-4.14|-3.21|-2.14|-0.81|-3.1|-0.37|-1.5|-1.2|-2.59|
> |$\Delta$ Avg. Other Domains|-0.42|-0.05|-0.28|-0.51|-0.08|-0.06|0.04|-0.21|-0.2|0.03|-0.02|-0.24|-0.28|-0.21|
> |$\Delta$ Std. Other Domains|0.52|0.9|0.93|0.74|0.69|0.66|0.67|0.57|0.66|0.79|0.7|0.71|0.81|0.61|
>
> - Mean of $\Delta$ Target: -2.65
> - Mean of $\Delta$ Avg. Other: -0.18
> - Mean of $\Delta$ Std. Other: 0.71
>
> **OLMoE 6.9B’s Domain Masking Performance Perturbation in MMLU**
>
> ||biology|business|chemistry|compsci|economics|engineering|health|history|law|math|other|philosophy|physics|psychology|
> |-|:-:|:-:|:-:|:-:|:-:|:-:|:-:|:-:|:-:|:-:|:-:|:-:|:-:|:-:|
> |$\Delta$ Target Domain|-1.74|-5.89|-4.46|-9.47|-3.68|-6.9|-4.55|-8.62|-7.98|-6.56|-0.62|-4.74|-2.72|-0.86|
> |$\Delta$ Avg. Other Domains|-1.33|-2.86|-3.08|-0.4|-1.51|-4.29|-1.67|-3.8|-5|-3.22|-0.27|-1.91|-0.96|-0.66|
> |$\Delta$ Std. Other Domains|1.3|1.78|2.04|1.18|1.62|2.38|2.08|2.15|2.22|2.51|1.11|1.49|1.55|0.68|
>
> - Mean of $\Delta$ Target: -4.91
> - Mean of $\Delta$ Avg. Other: -2.21
> - Mean of $\Delta$ Std. Other: 1.72
>
> Our additional experimentations suggest that OLMoE may be constituted with polysemantic experts. Results can be summarized as the following:
>
> 1. In OLMoE, there were extremely few specialized experts for MMLU, based on our criteria of skewness in expert routing score. In the case of Monet, we identified specialized experts if its highest routing score on a particular domain is twice as much as that of the second highest domain. However, OLMoE’s experts’ routing score was evenly distributed, making it difficult to detect specialized experts. We leveraged criteria of occurrences in maximum activation to determine the expert’s domain specialization to obtain the results.
> 2. OLMoE’s accuracy drop in other domains was significant in unlearning, possibly due to the entangled characteristics of experts since their specializations were only detectable with argmax criteria.
> 3. We measured delta performances’ mean standard deviation of the other 13 domains, resulting in 0.7 for Monet and 1.7 for OLMoE, differing twice as much, showing disparity in stability of knowledge conservation during unlearning.
>
> We believe that such results suggest that for most of the SMoE architectures [3-6] with 64 experts or less, the expert count is too small to disentangle polysemanticity. Our architecture, on the other hand, has 262,144 experts available, which we believe enable fine-grained specialization, resulting in monosemantic experts that capture mutually exclusive aspects of knowledge. To further address your inquiry, we provide an overview of the unlearning results of Monet, Gemma Scope, OLMoE, and LLaMa in `Figure 3` in our revised paper.
>
> Despite the fact we have previously compared the time complexity and space complexity with the PEER baseline, we remind you that additional 100B parameters are needed to constitute a PEER baseline, as we have explained in `Section 2` of our paper. Such exorbitant memory requirements are beyond the scope of most researchers (note that PEER was introduced by Google Deepmind), where our contribution is to achieve parameter efficiency, **precisely because directly implementing the PEER baseline is infeasible**.

---

> > ### Comment · Reviewer_YJRi · 2024-11-23
> > **Re: Comparison with baseline**
> >
> > This doesn't answer my question fully: how this architecture choice fares against traditional architecture in apple to apple comparison in terms of final quality of model.
> >
> > The perspective on interpretability is interesting and I get the point. But it's not my main concern. As practitioner, this is a very important question to be answered.

---

> ### Author Response · Authors · 2024-11-21
> **Response to Reviewer YJRi (Part 3/3)**
>
> > Can we mix Horizontal Expert Decomposition and vertical expert decomposition?
> >
>
> Thank you for your suggestions on additional experiments where two orthogonal decomposition methods can be mixed and complement each other. The results are presented as below:
>
> **Summary of 8 open-ended LLM benchmarks**
> |  | Avg. Performance (0-shot) | Avg. Performance (5-shot) |
> | :---: | :---: | :---: |
> | Horizontal Decomposition (HD) | 0.463 | 0.487 |
> | Vertical Decomposition (VD) | 0.478 | 0.510 |
> | Complementary Mix (HD + VD) | 0.470 | 0.503 |
>
>
> **Details of 8 open-ended LLM benchmarks**
> | | MMLU | ARC | WG | PIQA | SIQA | OBQA | HellaSwag | CSQA | Avg |
> | :---: | :---: | :---: | :---: | :---: | :---: | :---: | :---: | :---: | :---: |
> | 0-shot |  |  |  |  |  |  |  |  |  |
> | Horizontal Decomposition (HD) | 0.338 | 0.471 | 0.538 | 0.714 | 0.418 | 0.382 | 0.501 | 0.339 | 0.463 |
> | Vertical Decomposition (VD) | 0.352 | 0.495 | 0.522 | 0.727 | 0.423 | 0.418 | 0.529 | 0.363 | 0.478 |
> | Complementary Mix (HD + VD) | 0.338 | 0.504 | 0.541 | 0.726 | 0.403 | 0.382 | 0.521 | 0.349 | 0.470 |
> | 5-shot |  |  |  |  |  |  |  |  |  |
> | Horizontal Decomposition (HD) | 0.352 | 0.544 | 0.530 | 0.720 | 0.432 | 0.360 | 0.518 | 0.441 | 0.487 |
> | Vertical Decomposition (VD) | 0.360 | 0.547 | 0.526 | 0.730 | 0.441 | 0.422 | 0.551 | 0.501 | 0.510 |
> | Complementary Mix (HD + VD) | 0.355 | 0.567 | 0.541 | 0.717 | 0.437 | 0.384 | 0.537 | 0.489 | 0.503 |
>
> > **Q1**. citation to PEER is missing.\
> **Q2**. No proper ablations to study different choices in the architectural design and no insight is provided. \
> **Q3**. What does the model start with in table 3?
> >
>
> Respectfully, we would like to correct that
>
> - **A1**. A citation to PEER was already present in our `1. Introduction` section.
> - **A2**. An ablation study on auxiliary loss weights was present in `Appendix Section C.1`, where orthogonal architectural design choices have been rigorously compared in `Section 3`  across model sizes and benchmarks.
> - **A3**. `Table 3`’s full performance was also present in `Appendix Section E`.
>
> We understand that such a misconception is due to a lack of space in the paper where a fraction of the information had to be moved to the appendix. We would graciously ask you to read our revised manuscript if you could spare your invaluable time. Thank you once again.
>
> [1] Guillaume Lample, Alexandre Sablayrolles, Marc’Aurelio Ranzato, Ludovic Denoyer, and Hervé Jégou. Large Memory Layers with Product Keys. In Advances in Neural Information Processing Systems, volume 32, 2019.\
> [2] Xu Owen He. Mixture of a million experts. arXiv preprint arXiv:2407.04153, 2024\
> [3] Niklas Muennighoff, Luca Soldaini, Dirk Groeneveld, Kyle Lo, Jacob Morrison, et al. OLMoE: Open Mixture-of-Experts Language Models. arXiv preprint arXiv:2409.02060, 2024.\
> [4] Yikang Shen, Zhen Guo, Tianle Cai, and Zengyi Qin. JetMoE: Reaching Llama2 Performance with 0.1M Dollars. arXiv preprint arXiv:2404.07413, 2024.\
> [5] Damai Dai, Chengqi Deng, Chenggang Zhao, R.x. Xu, Huazuo Gao, et al. DeepSeekMoE: Towards Ultimate Expert Specialization in Mixture-of-Experts Language Models. In Proceedings of the 62nd Annual Meeting of the Association for Computational Linguistics (Volume 1: Long Papers), pp. 1280–1297, August 2024.\
> [6] Jan Ludziejewski, Jakub Krajewski, Kamil Adamczewski, Maciej Pióro, Michał Krutul, et al. Scaling Laws for Fine-Grained Mixture of Experts. In ICLR 2024 Workshop on Mathematical and Empirical Understanding of Foundation Models, 2024.

---

> ### Author Response · Authors · 2024-11-25
> **Baseline Comparisons**
>
> Dear Reviewer @YJRi,
>
> Thank you for your insightful feedback. We understand that your primary concern is how our Monet architecture compares to traditional SMoE architectures in terms of the final quality of the model. As practitioners, we agree that assessing model performance is crucial alongside interpretability.
>
> To address your concern, we conducted additional experiments to provide a direct comparison between Monet and the state-of-the-art SMoE architecture, OLMoE [1]. We ensured a fair evaluation by matching both the number of active parameters and the total number of parameters, as well as training both models on the same amount of data.
>
> ## **Total Parameter Matched Comparison**
>
> In this setup, both models have a similar total parameter count and are trained on 100 billion tokens.
>
> ### **Overall Performance**
>
> |Model|#Total Params|#Tokens Trained|Zero-shot Avg.|5-shot Avg.|
> |:-:|:-:|:-:|:-:|:-:|
> |**Monet (Ours)**|4.1B|100B|**0.511**|**0.550**|
> |OLMoE|6.9B|100B|0.502|0.534|
>
> ### **Benchmark Results**
>
> **Zero-shot Performance**
>
> |Task|MMLU|ARC|WinoGrande|PIQA|SocialIQA|OBQA|HellaSwag|CommonsenseQA|Avg.|
> |:-:|:-:|:-:|:-:|:-:|:-:|:-:|:-:|:-:|:-:|
> |**Monet**|**0.380**|**0.547**|**0.557**|0.751|**0.437**|**0.424**|0.604|0.389|**0.511**|
> |OLMoE|0.349|0.521|0.551|**0.754**|0.432|0.384|**0.620**|**0.402**|0.502|
>
> **5-shot Performance**
>
> |Task|MMLU|ARC|WinoGrande|PIQA|SocialIQA|OBQA|HellaSwag|CommonsenseQA|Avg.|
> |:-:|:-:|:-:|:-:|:-:|:-:|:-:|:-:|:-:|:-:|
> |**Monet**|**0.398**|**0.625**|**0.564**|**0.761**|**0.470**|**0.438**|0.619|0.525|**0.550**|
> |OLMoE|0.359|0.542|0.555|0.757|0.453|0.410|**0.637**|**0.561**|0.534|
>
> ## **Active Parameter Matched Comparison**
>
> To ensure an apples-to-apples comparison within our limited time frame, we conducted the active parameter matched experiments over a shorter training period. Both models have the same number of active parameters (1.3B) and were trained on 20 billion tokens.
>
> ### **Overall Performance**
>
> |Model|#Active Params|#Tokens Trained|Zero-shot Avg.|5-shot Avg.|
> |:-:|:-:|:-:|:-:|:-:|
> |**Monet (Ours)**|1.3B|20B|**0.457**|**0.479**|
> |OLMoE|1.3B|20B|0.432|0.453|
>
> ### **Benchmark Results**
>
> **Zero-shot Performance**
>
> |Task|MMLU|ARC|WinoGrande|PIQA|SocialIQA|OBQA|HellaSwag|CommonsenseQA|Avg.|
> |:-:|:-:|:-:|:-:|:-:|:-:|:-:|:-:|:-:|:-:|
> |**Monet**|**0.327**|**0.473**|**0.533**|**0.711**|0.418|**0.368**|**0.490**|0.338|**0.457**|
> |OLMoE|0.298|0.405|0.513|0.697|**0.421**|0.334|0.447|**0.343**|0.432|
>
> **5-shot Performance**
>
> |Task|MMLU|ARC|WinoGrande|PIQA|SocialIQA|OBQA|HellaSwag|CommonsenseQA|Avg.|
> |:-:|:-:|:-:|:-:|:-:|:-:|:-:|:-:|:-:|:-:|
> |**Monet**|**0.334**|**0.531**|**0.521**|**0.703**|**0.437**|**0.356**|**0.502**|**0.449**|**0.479**|
> |OLMoE|0.306|0.454|0.517|0.694|0.432|0.316|0.463|0.441|0.453|
>
> ### **Discussion**
>
> The results indicate that Monet consistently outperforms the traditional SMoE model across multiple benchmarks in both zero-shot and 5-shot settings. By matching both the total and active parameter counts, we ensured that the performance gains are attributable to the architectural differences rather than model size or training data volume. **These findings demonstrate that Monet not only offers improved interpretability but also delivers superior performance compared to conventional SMoE architectures.**
>
> We have revised the manuscript accordingly to include these comparisons and address your feedback. We appreciate your suggestion, as it encouraged us to perform this comprehensive comparison. We hope this addresses your concern regarding the final quality of the model. Please let us know if you have any further questions or suggestions.
>
> [1] Niklas Muennighoff, Luca Soldaini, Dirk Groeneveld, Kyle Lo, Jacob Morrison, et al. OLMoE: Open Mixture-of-Experts Language Models. arXiv preprint arXiv:2409.02060, 2024.

---

> > ### Comment · Reviewer_YJRi · 2024-11-25
> > **Thank you for the effort on showing proper controlled experiments**
> >
> > This is exactly what i need to see to be convinced. Best of luck!

---

> > > ### Author Response · Authors · 2024-11-26
> > > **Thank you for your valuable feedback**
> > >
> > > We appreciate your support and are grateful for your endorsement of our paper’s acceptance. Thank you.

---

### Official Review · Reviewer_sHPn · 2024-11-01

**Soundness:** 3
**Presentation:** 2
**Contribution:** 3
**Rating:** 6
**Confidence:** 3

**Summary:**

This paper presents a new architecture that makes large language models more interpretable with monosemanticity. The authors develop novel decomposition methods to efficiently scale to 262K experts per layer, achieving specialists that focus on single concepts through end-to-end training. The model also enables control over model knowledge (across domains, languages, and toxicity) without degrading performance, outperforming traditional Sparse Autoencoder approaches.

**Strengths:**

- The paper presents novel decomposition methods that scales traditional MoE to 262k experts.
- The paper delivers comprehensive experimental results on the proposed model architecture.
- The proposed method achieves good expert specialization, proven under several experimental settings

**Weaknesses:**

- The intuition behind the architecture design is unclear.
- The explanation in the methodology section is poor and hard to understand.

**Questions:**

- What's the reason of choosing $512^2$ number of experts?
- Are there any trade-offs for adopting Monet over traditional MoE? What is the training time comparison between Monet and LLaMA baseline models?

I suggest the authors to elaborate more on the methodology section.

**Details Of Ethics Concerns:**

No ethcis concerns are needed for the paper.

---

> ### Author Response · Authors · 2024-11-21
> **Response to Reviewer sHPn**
>
> We would like to express our gratitude to the reviewer for their constructive response. Below we respond to the weaknesses and questions.
>
> > **Q1**. The explanation in the methodology section is poor and hard to understand.\
> **Q2**. Are there any trade-offs for adopting Monet over traditional MoE?\
> **Q3**. The intuition behind the architecture design is unclear.\
> **Q4**. What's the reason of choosing $512^2$ experts?
> >
>
> **Please refer to our updated manuscript**, where we have improved the readability in sections `2. Preliminaries` to through `3. Monet`. We appreciate your comments about the clarity, and we are sorry about the confusion.
>
> To summarize sections 2 and 3:
>
> 1. Inspired by the product key algorithm [1], PEER [2] processes up to a million experts with product key retrieval.
> 2. Despite its computational efficiency, PEER requires to initialize and store $N$ standalone experts, resulting in memory usage that grows linearly with the number of experts, $O(N)$.
> 3. In response to **Q1**, our contribution is partitioning the expert’s MLP network into two different groups of segments and storing them within $O(\sqrt{N})$ memory constraint. During the training or inference, the learned router dynamically composes expert networks to form $N$ combinations of experts.
>
> Below is a comparison of time complexity for expert retrieval and space complexity for expert parameters to address **Q2**:
>
> | **Model** | **Time Complexity** | **Space Complexity** |
> | --- | --- | --- |
> | SMoE | $O(Nd)$ | $O(Nmd)$ |
> | PEER | $O((\sqrt{N} + k^2)Hd)$ | $O(Nd)$ |
> | Monet | $O(\sqrt{N}Hd)$ | **$O(\sqrt{N}md)$** |
>
> where $d$ is the hidden dimension of the expert, $m$ is the dimension of the individual expert, $k$ is the TopK hyperparameter, and $H$ denotes multi-head of the router. Individual expert dimension $m$ can be any value in our architecture, but PEER had to use a fixed value of $m=1$ because of a memory bottleneck.
>
> - Regarding **Q3**, our purpose is to optimize space complexity while maintaining the time complexity of PEER.
> - For **Q4**, we have followed the product key counts as mentioned in [1] of $512^2$ for our product key composition.
>
> Thank you for your thoughtful feedback that has helped refine our paper. We welcome any further questions or suggestions that could enhance the contribution of our work to the field.
>
> [1] Guillaume Lample, Alexandre Sablayrolles, Marc’Aurelio Ranzato, Ludovic Denoyer, and Hervé Jégou. Large Memory Layers with Product Keys. In Advances in Neural Information Processing Systems, volume 32, 2019.\
> [2] Xu Owen He. Mixture of a million experts. arXiv preprint arXiv:2407.04153, 2024.

---

> > ### Comment · Reviewer_sHPn · 2024-11-26
> > **Official Comment by Reviewer sHPn**
> >
> > I appreciate the authors for the rebuttal. I recommend the acceptance of the paper due to the strong novelty of the work.

---

> > > ### Author Response · Authors · 2024-11-26
> > >
> > > Dear Reviewer sHPn,
> > >
> > > Thank you for your support and for recommending the acceptance of our paper due to its strong novelty. We are grateful for your positive feedback.
> > >
> > > We notice that your overall rating remains unchanged. If there are any remaining concerns or suggestions you have for improving our submission, we would greatly appreciate your guidance. Your insights are valuable to us, and we are committed to addressing any outstanding issues.

---

> > > ### Author Response · Authors · 2024-12-01
> > >
> > > Dear Reviewer sHPn,
> > >
> > > Thank you for your thoughtful feedback and for recommending the acceptance of our paper due to its strong novelty. We are glad that our responses have addressed your main concerns.
> > >
> > > As the author-reviewer discussion period is nearing its end, we wanted to inquire if there are any remaining questions or suggestions you might have. If our revisions have satisfactorily addressed your concerns, we kindly ask you to consider reflecting this in your final evaluation.
> > >
> > > We sincerely appreciate your time and contributions to improving our work. Please feel free to share any additional feedback, and we will be more than happy to discuss and incorporate it.
> > >
> > > Thank you once again for your support.
> > >
> > > Best regards,
> > >
> > > The Authors.

---

### Official Review · Reviewer_53hd · 2024-11-03

**Soundness:** 3
**Presentation:** 2
**Contribution:** 3
**Rating:** 8
**Confidence:** 4

**Summary:**

This paper introduces the use of Mixture of Experts as a way to have more interpretable models in the context of polysemanticity. They change the standard MoE architecture in that they use product key retrieval technique as a router and they have experts associated with each key. They consider two strategies to create the model: horizontal expert decomposition and vertical expert decomposition, and finally explain how to train their models (Section 3). In the experiments section (Section 4), they show that the experts display monosemanticity and that removing some experts from some domain yields significant performance degradation (Sections 5.1 and 5.2). The Monet approach also allows to purge toxic experts from the model, which is interesting from a safety perspective.

**Strengths:**

I like the idea of the paper. Some earlier works noticed that experts display some monosemanticity [1,2] and it is great to see this work push this idea. I also think that the set of experiments is very convincing and I believe that this work may be influential for getting more interpretable neural networks.

[1] Fedus, William, Barret Zoph, and Noam Shazeer. "Switch transformers: Scaling to trillion parameter models with simple and efficient sparsity." Journal of Machine Learning Research 23.120 (2022): 1-39.

[2] Fedus, William, Jeff Dean, and Barret Zoph. "A review of sparse expert models in deep learning." arXiv preprint arXiv:2209.01667 (2022).

**Weaknesses:**

I think the main weakness of the paper is the presentation + writing, especially in Section 3. I am happy to consider improving my score if much better explanations of the method are given in Section 3.

- **Section 3 should be more clear (especially the Horizontal and Vertical decomposition)**: I read the work by Lample et al. [1] for completing this review and according to my understanding, there is a unique $(u_i, v_i)$ that is associated with each key. Their approach makes sense to me.

  -- I am very confused why there is the mix and match (along the horizontal or the vertical) in this paper. And also, why is there any memory savings (compared to the PEER approach)? And why is each expert of dimension m (while in PEER, it is a single neuron).


  -- I also recommend the authors to do a complexity calculation like in [1], Section 3.2 to be fully transparent on the memory/computation complexities.

  -- I also didn’t find Figure 1 very clear, for instance it was not clear what “Top”, “bottom” or “TL”, “BL” refer to. Above all, I think that this drawing should be improved.


- **Lack of baselines**: It is also not clear to me that a whole new architecture is needed to ensure a more interpretable model. For instance, [2,3] showed that standard MoEs display monosemanticity behaviors. Therefore, I think it is important to maybe compare the Monet method with standard MoEs. Would for instance fine-grained MoEs [4] work in this case? Is it the fact that we have a lot of experts that is responsible for more “monosemantic” experts? Or the routing strategy is responsible for it? I just want to be convinced that no simpler architecture would lead to the results obtained in Section 4.



[1] Lample, Guillaume, Alexandre Sablayrolles, Marc'Aurelio Ranzato, Ludovic Denoyer, and Hervé Jégou. "Large memory layers with product keys." Advances in Neural Information Processing Systems 32 (2019).

[2] Fedus, William, Barret Zoph, and Noam Shazeer. "Switch transformers: Scaling to trillion parameter models with simple and efficient sparsity." Journal of Machine Learning Research 23.120 (2022): 1-39.

[3] Fedus, William, Jeff Dean, and Barret Zoph. "A review of sparse expert models in deep learning." arXiv preprint arXiv:2209.01667 (2022).

[4] Krajewski, Jakub, Jan Ludziejewski, Kamil Adamczewski, Maciej Pióro, Michał Krutul, Szymon Antoniak, Kamil Ciebiera et al. "Scaling laws for fine-grained mixture of experts." arXiv preprint arXiv:2402.07871 (2024).

**Questions:**

I listed my question in the weaknesses section.

---

> ### Author Response · Authors · 2024-11-21
> **Response to Reviewer 53hd (Part 1/2)**
>
> We would like to express our gratitude for your positive feedback on our paper's idea and the effort you invested in its assessment. In the following response, we will address each of the weaknesses and questions you have raised.
>
> > Section 3 should be more clear.\
> **Q1**. Why is there any memory savings (compared to the PEER approach)?\
> **Q2**. Why is each expert of dimension m (while in PEER, it is a single neuron)?\
> **Q3**. I also recommend the authors to do a complexity calculation, Section 3.2 to be fully transparent on the memory/computation complexities.\
> **Q4**.  I think that this drawing should be improved.
> >
>
> **Please refer to our updated manuscript**, where we have improved the readability in sections `2. Preliminaries` to through `3. Monet`. We appreciate your comments about the clarity, and we are sorry about the confusion.
>
> To summarize sections 2 and 3:
>
> 1. Inspired by the product key algorithm [1], PEER [2] processes up to a million experts with product key retrieval.
> 2. Despite its computational efficiency, PEER requires to initialize and store $N$ standalone experts, resulting in memory usage that grows linearly with the number of experts, $O(N)$.
> 3. In response to **Q1**, our contribution is partitioning the expert’s MLP network into two different groups of segments and storing them within $O(\sqrt{N})$ memory constraint. During the training or inference, the learned router dynamically composes expert networks to form $N$ combinations of experts.
>
> Below is a comparison of time complexity for expert retrieval and space complexity for expert parameters:
>
> | **Model** | **Time Complexity** | **Space Complexity** |
> | --- | --- | --- |
> | SMoE | $O(Nd)$ | $O(Nmd)$ |
> | PEER | $O((\sqrt{N} + k^2)Hd)$ | $O(Nd)$ |
> | Monet | $O(\sqrt{N}Hd)$ | **$O(\sqrt{N}md)$** |
>
> where $d$ is the hidden dimension of the expert, $m$ is the dimension of the individual expert, $k$ is the TopK hyperparameter, and $H$ denotes multi-head of the router.
>
> - Regarding **Q2**, dimension $m$ can be any value in our architecture, but PEER had to use a fixed value of $m=1$ because of a memory bottleneck.
> - Regarding **Q3**, specific complexity calculation is present in `Appendix A.2` in our updated manuscript, where the table above provides a brief overview and comparison.
> - Based on your feedback **Q4**, we have also enhanced the presentations of `Figure 1` in the revision.
>
> [1] Guillaume Lample, Alexandre Sablayrolles, Marc’Aurelio Ranzato, Ludovic Denoyer, and Hervé Jégou. Large Memory Layers with Product Keys. In Advances in Neural Information Processing Systems, volume 32, 2019.\
> [2] Xu Owen He. Mixture of a million experts. arXiv preprint arXiv:2407.04153, 2024.

---

> ### Author Response · Authors · 2024-11-21
> **Response to Reviewer 53hd (Part 2/2)**
>
> > Lack of baselines\
> **Q1**. I think it is important to maybe compare the Monet method with standard MoEs\
> **Q2**. Would for instance fine-grained MoEs work in this case?\
> **Q3**. Is it the fact that we have a lot of experts that is responsible for more “monosemantic” experts?
> >
>
> Following your request of a fine-grained SMoE interpretability baseline in **Q1** and **Q2**, we have included knowledge unlearning of OLMoE [3]. OLMoE LLM with total 6.9B parameters has been selected as the representative baseline for conventional SMoE architectures for two reasons: (1) it has the largest number of experts among the publicly available SMoE LLMs [3-5] and (2) it has been trained with an extensive amount of tokens from various sources.
>
> **Monet-VD 1.4B’s Domain Masking Performance Perturbation in MMLU**
>
> ||biology|business|chemistry|compsci|economics|engineering|health|history|law|math|other|philosophy|physics|psychology|
> |-|:-:|:-:|:-:|:-:|:-:|:-:|:-:|:-:|:-:|:-:|:-:|:-:|:-:|:-:|
> |$\Delta$ Target Domain|-4.66|-4.61|-5.49|-1.05|-2.32|-4.14|-3.21|-2.14|-0.81|-3.1|-0.37|-1.5|-1.2|-2.59|
> |$\Delta$ Avg. Other Domains|-0.42|-0.05|-0.28|-0.51|-0.08|-0.06|0.04|-0.21|-0.2|0.03|-0.02|-0.24|-0.28|-0.21|
> |$\Delta$ Std. Other Domains|0.52|0.9|0.93|0.74|0.69|0.66|0.67|0.57|0.66|0.79|0.7|0.71|0.81|0.61|
>
> - Mean of $\Delta$ Target: -2.65
> - Mean of $\Delta$ Avg. Other: -0.18
> - Mean of $\Delta$ Std. Other: 0.71
>
> **OLMoE 6.9B’s Domain Masking Performance Perturbation in MMLU**
>
> ||biology|business|chemistry|compsci|economics|engineering|health|history|law|math|other|philosophy|physics|psychology|
> |-|:-:|:-:|:-:|:-:|:-:|:-:|:-:|:-:|:-:|:-:|:-:|:-:|:-:|:-:|
> |$\Delta$ Target Domain|-1.74|-5.89|-4.46|-9.47|-3.68|-6.9|-4.55|-8.62|-7.98|-6.56|-0.62|-4.74|-2.72|-0.86|
> |$\Delta$ Avg. Other Domains|-1.33|-2.86|-3.08|-0.4|-1.51|-4.29|-1.67|-3.8|-5|-3.22|-0.27|-1.91|-0.96|-0.66|
> |$\Delta$ Std. Other Domains|1.3|1.78|2.04|1.18|1.62|2.38|2.08|2.15|2.22|2.51|1.11|1.49|1.55|0.68|
>
> - Mean of $\Delta$ Target: -4.91
> - Mean of $\Delta$ Avg. Other: -2.21
> - Mean of $\Delta$ Std. Other: 1.72
>
> Our additional experimentations suggest that OLMoE may be constituted with polysemantic experts. Results can be summarized as the following:
>
> 1. In OLMoE, there were extremely few specialized experts for MMLU, based on our criteria of skewness in expert routing score. In the case of Monet, we identified specialized experts if its highest routing score on a particular domain is twice as much as that of the second highest domain. However, OLMoE’s experts’ routing score was evenly distributed, making it difficult to detect specialized experts. We leveraged criteria of occurrences in maximum activation to determine the expert’s domain specialization to obtain the results.
> 2. OLMoE’s accuracy drop in other domains was significant in unlearning, possibly due to the entangled characteristics of experts since their specializations were only detectable with argmax criteria.
> 3. We measured delta performances’ mean standard deviation of the other 13 domains, resulting in 0.7 for Monet and 1.7 for OLMoE, differing twice as much, showing disparity in stability of knowledge conservation during unlearning.
>
> We believe that such results suggest that for most of the SMoE architectures [3-6] with 64 experts or less, the expert count is too small to disentangle polysemanticity. Our architecture, on the other hand, has 262,144 experts available, which we believe enable fine-grained specialization, resulting in monosemantic experts that capture mutually exclusive aspects of knowledge. To further address your inquiry of **Q3**, we provide an overview of unlearning results of Monet, Gemma Scope, OLMoE, and LLaMa in `Figure 3` in our revised paper.
>
> We sincerely appreciate your thorough review and valuable suggestions, which have helped strengthen our manuscript substantially. We remain available to address any additional questions or concerns you may have.
>
> [3] Niklas Muennighoff, Luca Soldaini, Dirk Groeneveld, Kyle Lo, Jacob Morrison, et al. OLMoE: Open Mixture-of-Experts Language Models. arXiv preprint arXiv:2409.02060, 2024.\
> [4] Yikang Shen, Zhen Guo, Tianle Cai, and Zengyi Qin. JetMoE: Reaching Llama2 Performance with 0.1M Dollars. arXiv preprint arXiv:2404.07413, 2024.\
> [5] Damai Dai, Chengqi Deng, Chenggang Zhao, R.x. Xu, Huazuo Gao, et al. DeepSeekMoE: Towards Ultimate Expert Specialization in Mixture-of-Experts Language Models. In Proceedings of the 62nd Annual Meeting of the Association for Computational Linguistics (Volume 1: Long Papers), pp. 1280–1297, August 2024.\
> [6] Jan Ludziejewski, Jakub Krajewski, Kamil Adamczewski, Maciej Pióro, Michał Krutul, et al. Scaling Laws for Fine-Grained Mixture of Experts. In ICLR 2024 Workshop on Mathematical and Empirical Understanding of Foundation Models, 2024.

---

### Official Review · Reviewer_oJuH · 2024-11-04

**Soundness:** 3
**Presentation:** 3
**Contribution:** 3
**Rating:** 8
**Confidence:** 4

**Summary:**

The paper proposes a new transformer architecture that replaces MLP layers in the standard decoder-only transformer architecture with a type of sparse coding layer which encourages only a small number of hidden neurons to activate on each given input. The construction is also motivated by, and borrows ideas from, the mixture of experts (MoE) literature. The primary motivation of this new architecture is to help interpretability by building something akin to a wide Sparse Autoencoder (SAE) into the MLP layers of the decoder-only transformer architecture in a scalable way, so that we can directly train for sparse (and thus hopefully interpretable) internal activations.

In more detail:
- the MLP layer is viewed as an associative memory, and replaced with a sparsely activating version inspired by the paper "Large memory layers with product keys".
	- The MLP layer is replaced by multiple smaller MLP subnetworks ("experts") that share parameters in a specific way inspired by the product idea from "Large memory layers with product keys" to effectively represent many experts using only a few trainable parameters.
	- A sparse subset of the experts is chosen to produce the final output as an expectation over these layers' outputs (similar to attention)
	- There are other engineering optimizations used to make the computation more efficient.
	- Finally, auxiliary loss terms are added, encouraging the experts to activate uniformly on average ("load balancing") and each token to have a highly activating expert (ambiguity loss).
- This new architecture is trained on 100B tokens sampled from the FineWeb-Edu dataset (a subset of experiments also uses a programming dataset), using LLaMA trained on the same dataset as a baseline across approx. 850M, 1.4B and 4.1B parameters. The MONET architecture uses an effective count of $2^18=262,144$ experts. Comparisons on question-answering benchmarks such as MMLU show that the architecture performs mostly on par with the LLaMA baseline.
- As an additional baseline, SAEs for Gemma 2B are used to patch in Gemma-2B's original activations, and the performance drop due to the SAEs is measured.
- Some qualitative analyses of the contexts that activate a given expert subnetwork are performed.
- The architecture is then applied to selectively delete model knowledge in three setups: subject-specific knowledge in MMLU (e.g. delete only knowledge of chemistry but not economics etc.), programming language-specific knowledge on a code dataset (e.g. delete only knowledge of Python but not Java), and purging toxic experts.

**Strengths:**

- The paper tackles an interesting and important question for the field: instead of interpreting LLMs post-hoc, can we directly train them in a way that results in interpretable weights?
	- This adds to existing work, such as backpack LLMs https://arxiv.org/abs/2305.16765 and codebook features https://arxiv.org/abs/2310.17230
- The proposed architecture is interesting, can (in principle) represent a large number of experts, and performs on par with the LLaMA baseline of roughly the same parameter count.
- The applications to targeted erasure of knowledge are very interesting and relevant to the field.
- The writing is clear

**Weaknesses:**

- The lack of detailed interpretability baselines makes it difficult to evaluate the strength of the results.
	- For example, the only interpretability method used as a baseline is patching reconstructions from SAEs for Gemma-2B. However, it is not reported what sparsity these SAEs achieve compared to the (effective?) sparsity of MONET. This makes it difficult to make sense of the results.
	- The only relevant baseline here is using SAEs at the MLP layers, because this matches the MONET setup; so, the residual stream SAEs seem irrelevant for this work?
	- Furthermore, SAEs are trained to reconstruct activations coming from the original model being studied, and iteratively applying the SAE reconstructions to MLP layers may take downstream activations off-distribution, leading to an accumulation of errors due to SAE composition. You may argue that this is just a drawback of the SAE paradigm that MONET avoids, and the comparison is still fair. However, from my point of view, the primary goal of SAEs is to find interesting concepts used by the model, and reconstruction is secondary to that (and being able to chain SAE reconstructions is even more secondary). So, ideally the baseline would compare the "monosemanticity" of MONET features vs SAE ones.
	- A baseline using the ordinary MLP neurons of the LLaMA model would be very valuable to make the point that MONET discovers more interpretable structure compared to the neuron basis
- The paper would benefit from a discussion of, and comparison with, related work, such as backpack language models and codebook features.
- Perhaps adding extra bells and whistles like instruction tuning or multimodality distracts from the main goal of the paper, which is to establish the usefulness of the new architecture for interpretability (which I believe can be achieved or falsified in a more basic setup)

**Questions:**

- How exactly were the top experts by subdomain chosen for the Gemma-2B SAEs? Note that SAEs have no notion of probability over the "experts", unlike the MONET model, and I could not find this addressed in the paper. Do you pass the hidden SAE activations through a softmax first?
- What is the scale in figure 2?
- Have you tried running the MONET features through an automated interpretability pipeline like https://github.com/EleutherAI/sae-auto-interp?

---

> ### Author Response · Authors · 2024-11-21
> **Response to Reviewer oJuH (Part 1/2)**
>
> We sincerely thank the reviewer for your helpful and constructive suggestions. In the following response, we will explicate the changes that have been made to the manuscript and a new version uploaded.
>
> > Have you tried running the MONET features through an automated interpretability pipeline?
>
> We express gratitude for suggesting such valuable feedback, where we have reflected the changes in `Figure 2` and in the `4.3 Qualitative Results` section in our revised manuscript.
>
> The attached example, [sae-auto-interp](https://github.com/EleutherAI/sae-auto-interp) has its significance in generating explanations of SAE features via external LLM or through the compatible API. We agree that features within the model should be able to be described in natural languages, considering its significance in controlling and managing LLMs.
>
> Taking your advice, we decided to take a step further, referring to the Self-explaining SAE features [2]. It states that it has advantages over [sae-auto-interp](https://github.com/EleutherAI/sae-auto-interp) of the following: no max activating dataset examples are needed, and it’s cheaper by using the model subject of study to generate about its own features rather than relying on a larger model like GPT-4.
>
> **Without using external LLMs or APIs, we adapted an automated interpretation framework as `Self-explained Experts`**, where Monet-1.4B CHAT generates a description for its own experts. We have referred to the work of Patchscope [3] and SelfIE [4], where both works prompt LLM to answer “Q: What is the meaning of the word X? A: Sure! The meaning of the word X is ”, where X serves as a placeholder for target token embedding for the analyses. Similarly, we averaged token embeddings activated for the targeted expert and have inserted them into the aforementioned placeholder. Our Monet-1.4B CHAT generated a description for its experts, like explaining the Expert 232,717 as “Cartilage” and the Expert 51 as “Expertise”, as stated in our revised manuscript.
>
> > **Q1**. The paper would benefit from a discussion of, and comparison with, related work, such as backpack language models and codebook features. \
> **Q2**. Perhaps adding extra bells and whistles like instruction tuning or multimodality distracts from the main goal of the paper, which is to establish the usefulness of the new architecture for interpretability.
>
> We thank your opinion on the paper’s related works and its main goal.
>
> - We have reviewed Backpack LLMs [5] and Codebook Features [6] according to your advice **Q1**, where we found encoding interpretable weights in LLM during pretraining shares similar philosophy in achieving interpretable models. In our `1. Introduction` section, we have reflected the change accordingly.
> - Furthermore, we value your advice **Q2** and took out the examples of multimodal experts (`Figures 9 and 10`) from the main text and moved to the appendix section. The rationale for staying in the paper is that it is yet unknown whether it is generalizable for fine-grained experts to specialize in and capture monosemantic concepts across modalities with finetuning. We would appreciate it if you could reconsider the significance of analyzing the expandability of our method in LLM’s multimodal integration to remain in the paper’s appendix section.
> - In the case of instruction tuning, the process was a precursor of the automated interpretability pipeline. Adhering to your suggestion, we have excluded the specifics regarding instruction tuning and moved to the Appendix, but we discussed its role in `Self-explained Experts` as we mentioned above in the previous response.
>
> > A baseline using the ordinary MLP neurons of the LLaMA model would be very valuable to make the point that MONET discovers more interpretable structure compared to the neuron basis.
>
> Thank you for your insightful suggestion. In response, we have included the LLaMA unlearning baseline in `Figure 3` and in `5.1 Domain Masking` section of our revised manuscript.
>
> In our experiments, we suppressed domain-specific MLP neurons based on first-layer activations. Inspired by the ROME [1] treating MLP as key-value pairs, we identified neurons with domain specialization based on GELU activations. Specifically, if the highest activation of a particular domain is twice as much as that of the second highest activated domain, we consider that neuron a specialized neuron.
>
> For the results, LLaMA displays an average 6% of neurons to be specialized in each domain compared to Monet's 2.2%, suggesting possible feature entanglement and resulting in significant performance degradation across unrelated domains during knowledge removal. We measured delta performances’ mean standard deviation of the other 13 domains, resulting in 0.7 for Monet and 1.4 for LLaMa, differing twice as much in stability of knowledge conservation during unlearning. Such results highlight Monet’s monosemanticity, where experts encapsulate disentangled parametric knowledge across domains.

---

> ### Author Response · Authors · 2024-11-21
> **Response to Reviewer oJuH (Part 2/2)**
>
> > How exactly were the top experts by subdomain chosen for the Gemma-2B SAEs? Note that SAEs have no notion of probability over the "experts", unlike the MONET model, and I could not find this addressed in the paper. Do you pass the hidden SAE activations through a softmax first?
> >
>
> We referred to the steering methods with SAEs, such as clamping the feature activations [7, 8] based on their logit values. To adhere to the conventional logit-based steering, we analyzed the skewness of SAE’s logit values, where we determine the feature is specialized in the particular domain only when its highest logit value is at least twice higher than that of the second most activated domain.
>
> > The only relevant baseline here is using SAEs at the MLP layers, because this matches the MONET setup; so, the residual stream SAEs seem irrelevant for this work?
> >
>
> While SAEs at the MLP layers correspond to the Monet's fine-grained experts, we chose to include residual stream SAE results for comprehensiveness. The MLP-based comparisons demonstrate the core architectural benefits, while the residual stream results provide context within the broader landscape of interpretability research. This allows readers to evaluate Monet's effectiveness against both the most directly comparable baseline and current common practices in the field.
>
> > What is the scale in figure 2?
> >
>
> Regarding the scale and full performance of each Monet (ours), Gemma Scope, OLMoE, and LLaMa in MMLU domain unlearning, we have listed in `Appendix E`’s `Table 11` through `Table 14` for the specifics. Please refer to the revised manuscript, and if you have additional inquiries, we are happy to respond to further questions and comments.
>
> > • For example, the only interpretability method used as a baseline is patching reconstructions from SAEs for Gemma-2B. However, it is not reported what sparsity these SAEs achieve compared to the (effective?) sparsity of MONET. This makes it difficult to make sense of the results.  \
> • The primary goal of SAEs is to find interesting concepts used by the model, and reconstruction is secondary to that (and being able to chain SAE reconstructions is even more secondary). So, ideally the baseline would compare the "monosemanticity" of MONET features vs SAE ones.
> >
>
> We employed Gemma Scope with 262K features at $L_0 = 263$, its maximum provided sparsity setting. However, direct sparsity comparisons between Monet and SAE models are not methodologically sound due to fundamental architectural differences. While MoE models use top-k routing for sparse expert activation, this mechanism differs fundamentally from SAE's $L_0$ sparsity measure.
>
> Nevertheless, our Monet's theoretical sparsity would be $L_0$ is 512, derived from $|\mathcal{K}_h^1 \times \mathcal{K}_h^2| = 64$ across 8 multi-head routings. Despite this higher $L_0$ value, which traditionally would suggest lower monosemanticity, Monet achieves superior disjoint unlearning performance, as demonstrated in Figure 3 in our revised manuscript. This indicates that routing-based sparsity may be more effective at isolating and controlling specific knowledge domains compared to traditional SAE approaches.
>
> We thank you again for your constructive comments and for your efforts to improve the quality of our paper. Please let us know if you have any further questions or if we can provide further clarification.
>
> [1] Kevin Meng, David Bau, Alex Andonian, and Yonatan Belinkov. Locating and editing factual associations in GPT. In Advances in Neural Information Processing Systems, volume 35, pp. 17359–17372.\
> [2] Dmitrii Kharlapenko, neverix, Neel Nanda, and Arthur Conmy. Self-explaining SAE features. AI Alignment Forum, 2024. URL https://www.alignmentforum.org/posts/8ev6coxChSWcxCDy8 \
> [3] Asma Ghandeharioun, Avi Caciularu, Adam Pearce, Lucas Dixon, and Mor Geva. Patchscope: A Unifying Framework For Inspecting Hidden Representations of Language Models. arXiv preprint arXiv:2401.06102, 2024.\
> [4] Haozhe Chen, Carl Vondrick, and Chengzhi Mao. SelfIE: Self-Interpretation of Large Language Model Embeddings. arXiv preprint arXiv:2403.10949, 2024.\
> [5] John Hewitt, John Thickstun, Christopher D. Manning, and Percy Liang. Backpack Language Models. In Annual Meeting of the Association for Computational Linguistics, 2023.\
> [6] Alex Tamkin, Mohammad Taufeeque, and Noah D Goodman. Codebook Features: Sparse and Discrete Interpretability for Neural Networks. arXiv preprint arXiv:2310.17230, 2023\
> [7] Leo Gao, Tom Dupré la Tour, Henk Tillman, Gabriel Goh, Rajan Troll, Alec Radford, Ilya
> Sutskever, Jan Leike, and Jeffrey Wu. Scaling and evaluating sparse autoencoders. arXiv preprint
> arXiv:2406.04093, 2024.\
> [8] Adly Templeton*, Tom Conerly*, Jonathan Marcus, Jack Lindsey, Trenton Bricken, et al. Extracting Interpretable Features from Claude 3 Sonnet. Transformer Circuits Thread, 2024, URL https://transformer-circuits.pub/2024/scaling-monosemanticity

---

> > ### Author Response · Authors · 2024-11-27
> >
> > Dear Reviewer oJuH,
> >
> > Thank you for your thoughtful and constructive feedback on our manuscript. We have thoroughly addressed your comments and submitted a revised version for your consideration. We would greatly appreciate it if you could review our responses and the updated manuscript at your earliest convenience. We understand the demanding nature of the review process and are grateful for the time and effort you are dedicating to our work.

---

> > ### Comment · Reviewer_oJuH · 2024-11-27
> > **Response to rebuttals**
> >
> > Thank you for the detailed and thorough rebuttal. I think these new additions, especially the new baselines, improve the paper. I have a few remaining important concerns:
> > - the autointerpretability evaluation is exploratory and does not demonstrate metrics showing the improved interpretability of MONET compared to other methods;
> > - the results described in Figure 3 are very interesting, but they are a bit hard to read due to the unclear scale. In particular, looking at Appendix E, I found the last two rows of each table ($\Delta$ Target and $\Delta$ Others) to be most helpful in making sense of this dense data. I would advise somehow surfacing this in the camera ready if the paper is accepted;
> > - I feel that the exploration of methods for picking experts was insufficient. I would love to see future work/revisions more thoroughly tuning the choice of experts for each baseline as well as MONET.
> >
> > Thanks again to the authors for the very detailed and thorough response. I am raising my score as a result of these improvements.

---

> ### Author Response · Authors · 2024-11-28
> **Thank you for your valuable feedback**
>
> Thank you for your support and for endorsing the acceptance of our paper. In the following responses, we address each of the concerns you have raised and have updated the manuscript accordingly (*changes highlighted in magenta).
>
> > the autointerpretability evaluation is exploratory and does not demonstrate metrics showing the improved interpretability of MONET compared to other methods;
> >
>
> Thank you for raising this important point about quantitative evaluation of automated interpretability. **We want to clarify that primary objective of self-explained experts was to leverage LLMs' internal knowledge and capabilities by eliminating dependencies on external LLMs**, rather than to demonstrate superiority in automated interpretation frameworks. We note that concurrent work in quantitatively measuring automated interpretability of SAEs [1,2] are still in its early stage of development, which we view as an opportunity to develop more comprehensive evaluation protocols.
>
> While tools like `sae-auto-interp` provide valuable pipelines for generating and evaluating feature explanations, their quantitative evaluation frameworks are currently designed to compare different explanation methods between each LLM, rather than enabling direct comparisons between SAE models. We plan to prioritize developing more robust comparative frameworks in our future work to provide additional numerical assessment of Monet's automated interpretation framework.
>
> > the results described in Figure 3 are very interesting, but they are a bit hard to read due to the unclear scale. In particular, looking at Appendix E, I found the last two rows of each table (Δ Target and Δ Others) to be most helpful in making sense of this dense data. I would advise somehow surfacing this in the camera ready if the paper is accepted;
> >
>
> Thank you for your positive feedback on Figure 3. We appreciate your feedback that, while the results are very interesting, the unclear scale made them hard to read. We agree that relying on Appendix E for clarity might be inconvenient for readers. In response to your suggestion, **we have updated Figure 3 in the revised manuscript to include precise scales.** We believe these editorial changes will enhance the readability of our results. Thank you for bringing this to our attention, and we apologize for any inconvenience the original presentation may have caused.
>
> > I feel that the exploration of methods for picking experts was insufficient. I would love to see future work/revisions more thoroughly tuning the choice of experts for each baseline as well as MONET.
> >
>
> Thank you for your valuable feedback. We acknowledge that our exploration of methods for selecting experts was insufficient and agree that more thorough tuning is necessary.
>
> In our current work, we used the skewness of the routing score to determine experts' domain specialization and identified toxic experts using the Pearson correlation coefficient between the toxicity score and the routing score. We recognize that these criteria are basic and minimal.
>
> Our primary contribution lies in making the LLM transparent, enabling researchers to observe routing scores and directly manipulate the parametric knowledge. We believe that **the routing scores of monosemantic experts allow researchers to observe patterns for retrieving intrinsic knowledge, which were previously opaque in polysemantic LLMs.** We are optimistic that such observations can lead to addressing research questions related to hallucinations (e.g., "Is the model confident in retrieving internal knowledge?") and lifelong learning in LLMs (e.g., "How can we incorporate additional knowledge into the model?").
>
> **Based on your feedback, we have added a "Limitations" section to our paper**, summarizing the discussions above. Thank you once again for your insightful comments, which have been invaluable in guiding the future direction of our research.
>
> [1] Jack Lindsey, Hoagy Cunningham, and Tom Conerly. Interpretability Evals for Dictionary Learning. Transformer Circuits Thread, 2024, URL https://transformer-circuits.pub/2024/august-update/index.html#interp-evals
>
> [2] CHAUDHARY, Maheep; GEIGER, Atticus. Evaluating Open-Source Sparse Autoencoders on Disentangling Factual Knowledge in GPT-2 Small. arXiv preprint arXiv:2409.04478, 2024.

---

> > ### Comment · Reviewer_oJuH · 2024-11-28
> >
> > Thank you for the timely response and the quick revision of your work. These additions greatly improve the presentation of your work.
> >
> > Regarding the autointerpretability, I still think results that compare, using a single autointerpretability pipeline, the interpretability score of MONET experts versus SAE latents, but this seems like a better fit for future work, as it would also require developing a way to fairly compare your architecture to SAEs.

---

> > > ### Author Response · Authors · 2024-12-01
> > > **Future works on autointerpretability**
> > >
> > > Thank you for your encouraging feedback and for acknowledging the improvements in our revised manuscript. We are pleased to hear that the additions have enhanced the presentation of our work.
> > >
> > > **We concur that this endeavor is well-suited for future work, where we can dedicate effort to develop a robust and fair comparative methodology.** This would not only strengthen the evaluation of our model but also contribute to the broader research community by providing tools to assess interpretability across different architectures.
> > >
> > > Thank you once again for your insightful suggestions. We are committed to advancing this line of research and look forward to exploring these ideas in our future work.

---

### Author Response · Authors · 2024-11-21
**General Response to All Reviewers**

We sincerely appreciate the reviewers for their thoughtful and constructive feedback, which have greatly contributed to improving our work. We are pleased that the reviewers find our problem statement **important and interesting** (@oJuH), and believe our work may be **influential in the field of interpretable neural networks** (@53hd). Reviewers also consider our proposed architecture **novel and effective** (@oJuH, @53hd, @sHPn), and regard our **experiments as convincing and comprehensive** (@53hd, @sHPn).

In our responses to the reviews, we have carefully addressed all raised concerns. These can be summarized as follows:

- **Improved presentation and clarity**: We have enhanced the methods section and Figure 1 to facilitate a clearer understanding of our proposed product key composition. (@53hd, @sHPn, @YJRi)
- **Automated interpretation framework**: We have adapted an automated interpretation framework as Self-explained Experts without relying on external LLMs or APIs. This approach is discussed in Section 4.3, with results illustrated in Figure 2. (@oJuH)
- **Additional interpretability baselines of OLMoE and LLaMA**: We have incorporated additional interpretability baselines in Section 5.1 (Domain Masking) and illustrated them in Figure 3, where such baselines exhibited polysemanticity in unlearning. (@oJuH, @53hd, @YJRi)
- **Additional general performance comparisons**: We conducted additional experiments comparing Monet with the state-of-the-art SMoE architecture OLMoE under matched conditions, demonstrating Monet's superior performance across benchmarks. (@YJRi)
- **Complexity calculations**: We have included complexity calculations in Appendix A.2, demonstrating that our method efficiently reduces memory growth to $O(\sqrt{N}md)$, enabling us to scale the expert count to 262,144. (@53hd, @sHPn)

    | **Model** | **Time Complexity** | **Space Complexity** |
    | --- | --- | --- |
    | SMoE | $O(Nd)$ | $O(Nmd)$ |
    | PEER | $O((\sqrt{N} + k^2)Hd)$ | $O(Nd)$ |
    | Monet (Ours) | $O(\sqrt{N}Hd)$ | **$O(\sqrt{N}md)$** |

We have incorporated the feedback into our revised paper, highlighting the changes in blue for easy reference. Additional edits have been made to enhance clarity and conciseness. We welcome further questions or comments and will promptly address any concerns.

Thank you again,

The Authors.

---

### Meta-Review · Area_Chair_aByw · 2024-12-21

**Metareview:**

The reviewers commended the novel approach to embedding interpretability directly into large language models. By introducing sparse coding layers inspired by Mixture of Experts (MoE), the model achieves sparsity and interpretability without compromising performance. Reviewers highlighted its ability to selectively erase domain-specific knowledge, enhance safety, and enable practical applications, all while maintaining performance parity with LLaMA models on key benchmarks. The comprehensive experimental evaluations were widely praised, particularly MONET's robustness across diverse settings. The rebuttal addressed key concerns, added baselines, and clarified results.

**Additional Comments On Reviewer Discussion:**

The rebuttal addressed key concerns, added baselines, and clarified results.

---

### Decision · Program_Chairs · 2025-01-22

Accept (Poster)